# Detecting Aquaculture with Deep Learning in a Low-Data Setting

Laura Greenstreet
leg86@cornell.edu
Dept. of Computer Science, Cornell
University
Ithaca, NY, USA

Joshua Fan
jyf6@cornell.edu
Dept. of Computer Science, Cornell
University
Ithaca, NY, USA

Felipe Siqueira Pacheco
felipe.pacheco@cornell.edu
Dept. of Ecology and Evolutionary
Biology, Cornell University
Ithaca, NY, USA

Yiwei Bai
yb263@cornell.edu
Dept. of Computer Science, Cornell
University
Ithaca, NY, USA

Marta Eichemberger Ummus
marta.ummus@embrapa.br
Brazilian Agricultural Research
Corporation - (Embrapa)
Brasília, Brazil

Carolina Doria
carolinarcdoria@unir.br
Dept. of Biology, Federal University
of Rondônia
Porto Velho, Brazil

Nathan Oliveira Barros
nathan.barros@ufjf.edu.br
Dept. of Biology, Federal University
of Juiz de Fora
Juiz de Fora, Brazil

Bruce R. Forsberg
brforsberg@gmail.com
LBA Program Office, National
Institute of Amazon Research
Manaus, Brazil

Xiangtao Xu
xx268@cornell.edu
Dept. of Biology and Evolutionary
Ecology, Cornell University
Ithaca, NY, USA

Alexander Flecker
asf3@cornell.edu
Dept. of Ecology and Evolutionary
Biology, Cornell University
Ithaca, NY, USA

Carla Gomes
gomes@cs.cornell.edu
Dept. of Computer Science, Cornell
University
Ithaca, NY, USA

## ABSTRACT

Aquaculture is growing rapidly in the Amazon basin and detailed spatial information is needed to understand the trade-offs between food production, economic development, and environmental impacts. Large open-source datasets of medium resolution satellite imagery offer the potential for mapping a variety of infrastructure, including aquaculture ponds. However, there are many challenges utilizing this data, including few labelled examples, class imbalance, and spatial bias. We find previous rule-based methods for mapping aquaculture perform poorly in the Amazon. By incorporating temporal information through percentile data, we show deep learning models can outperform previous methods by as much as 15% with as few as 300 labelled examples. Further, generalization to unseen regions can be improved by incorporating segmentation information through masked pooling and using contrastive pretraining to harness large quantities of unlabelled data.

## CCS CONCEPTS

• **Computing methodologies** → **Object identification**; *Image segmentation*.

## KEYWORDS

remote sensing, image segmentation, image classification, attention, contrastive learning, representation learning, convolutional neural networks

**ACM Reference Format:**
Laura Greenstreet, Joshua Fan, Felipe Siqueira Pacheco, Yiwei Bai, Marta Eichemberger Ummus, Carolina Doria, Nathan Oliveira Barros, Bruce R. Forsberg, Xiangtao Xu, Alexander Flecker, and Carla Gomes. 2023. Detecting Aquaculture with Deep Learning in a Low-Data Setting. In *Proceedings of Make sure to enter the correct conference title from your rights confirmation email (SigKDD Fragile Earth Workshop)*. ACM, New York, NY, USA, 9 pages. https://doi.org/10.1145/nnnnnnn.nnnnnnn

## 1 INTRODUCTION

Aquaculture describes the farming or cultivation of aquatic organisms, including fish, molluscs, and crustaceans [7]. In contrast to capture fisheries, in aquaculture steps are taken to enhance production such as stocking, feeding, and controlling features of the aquatic environment. Aquaculture can help meet multiple Sustainable Development Goals, including reducing hunger, improving nutrition, and creating sustainable economic growth and has been identified as a key way to reduce the environmental impacts of animal-based foods [2, 41]. Previously, studies have shown that aquaculture generally has lower freshwater use, carbon emissions, and nutrient pollution than traditional livestock [9, 18]. However, impacts such as freshwater use and carbon emissions can vary by over two orders of magnitude based on factors such as species farmed, production methods, and land-use change [13]. Thus, information at the level of individual aquaculture farms is important

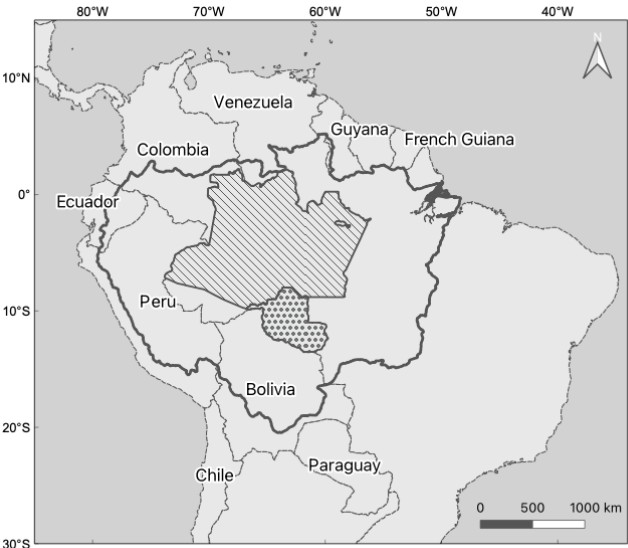

**Figure 1: The Amazon basin (solid outline) covers over 7 million square kilometers and intersects 8 countries. In this study, we train models for aquaculture detection using data from the Brazilian state of Rondônia (dotted region) and test generalization to the state of Amazonas (dashed region).**

to develop socially, economically, and environmentally sustainable food systems.

Currently, studies of aquaculture in the Amazon are hindered by a lack of comprehensive data for even basic information, such as farm location and pond sizes [8, 23]. Mapping aquaculture in the Amazon is particularly important due to the region's ecological significance. Detailed maps can assist in environmental impact assessments by revealing the spatial distribution of aquaculture activities and their proximity to vulnerable ecosystems [13, 22]. Additionally, detailed spatial information aids in identifying areas of potential expansion or conflict with other land uses [23, 42], and is instrumental in decision-making processes, allowing for informed policy creation and better management strategies [1]. Moreover, a comprehensive map also provides the foundation for monitoring and mitigating potential negative effects such as pollution, disease spread, or habitat degradation [24].

Large open-source datasets of medium-resolution remote sensing imagery create the possibility of large-scale, low-cost mapping of a variety of resources and infrastructure. For example, since 2014, the Sentinel-1 and 2 satellites have been producing a global dataset of both radar and multispectral data at 10-60m$^2$ resolution with a revisit time as low as 5 days [35]. However, there are several challenges to effectively using this data. First, while huge quantities of data are available, there are often few labelled examples. Second, labelled examples often have heavy class imbalances and are spatially biased, making it difficult to create reliable models that do not learn spurious correlations. Further, some resources such as crops or aquaculture ponds may only be distinguishable from

similar resources with time series imagery that captures patterns of human management over time.

## 1.1 Prior Work

Over eight papers have been published in the past five years mapping aquaculture from remote imagery [6, 10, 26, 28, 29, 31, 38, 44]. All studies focused on regions in Asia, ranging in scale from a small island [10] to the full continent [26]. The majority of these studies used Sentinel-1 and 2 medium resolution imagery (10-60m$^2$) as input, with Sentinel-1's synthetic aperture radar (SAR) playing an important role as it can penetrate the clouds that obscure optical imagery [26–29, 38, 44]. A few studies used very high resolution satellite imagery (VHRS, ≤3m$^2$) [10] or Landsat data (30m$^2$) [6, 31], due to the ability to handle high-resolution data when considering a small area or the need for Landsat's longer imaging record when considering land-use change respectively. All studies took a segment-then-classify approach, first segmenting water bodies using variants of connected component segmentation (CCS), and then using rule-based approaches, such as decision trees, for classification. These methods reported accuracies of 83-96%, though sensitivities (precision values) only ranged from 83-89%.

There are several reasons why these methods may not generalize well to the Amazon. First, all studies relied on expert opinion of high-resolution satellite imagery to validate classifications, which may have systematically misclassified some aquaculture ponds. For example, several studies reported that it was difficult for humans to distinguish aquaculture ponds from other man-made water bodies like salt pans and rice paddies [27, 31, 38]. Studies that tried to address this manually examined time series images to determine whether a water body was an aquaculture pond, salt pan, or rice paddy based on when and for how long it was drained or contained vegetation. Additionally, the majority of aquaculture farms in Asia are large complexes of geometric ponds, while aquaculture in the Amazon includes a wider range of pond sizes and shapes, such as ponds created by sectioning off parts of streams (see Fig. 2). These pond shapes may not be correctly classified by the hand-crafted features used in prior work.

Deep learning has the potential to address many of these limitations by eliminating the need for manual feature selection, making it easier to incorporate multispectral and temporal data. While to our knowledge no studies have previously used deep learning to map aquaculture, several studies have mapped other forms of infrastructure, including solar panels and livestock barns at national and even global scales [19, 25, 33]. To handle having only thousands of labelled examples, prior studies relied heavily on data augmentation. [33] used multiple images of the same location over time to further increase dataset size, using an unsupervised approach to remove images before operations were built. All three studies trained models with high recall and low precision, adding post-processing steps to filter out false positives.

While it is encouraging that previous studies have been able to successfully train deep models on satellite imagery with only thousands of examples, for some regions in the Amazon we only have hundreds of labeled examples. Further, we expect more heterogeneity in aquaculture ponds in the Amazon than in man-made objects such as livestock barns or solar panels. Further, previous

work suggests that temporal information is important for distinguishing aquaculture ponds from similar man-made water bodies such as salt pans or rice paddies. However, incorporating time series or multispectral information will increase the dimensionality of the data, increasing the number of model parameters that need to be fit with only a small number of examples.

In this work, we first demonstrate standard deep learning models can outperform the previous rule-based methods for classifying aquaculture in the Brazilian state of Rondônia using data from small subregions with only 300-400 labeled examples. We show performance can be further improved by incorporating temporal information through percentile data. Further, we test the models' ability to generalize to an unseen state with significantly different environmental conditions; we show that generalization can be improved by incorporating segmentation information through masked pooling methods and by leveraging large quantities of unlabeled data through contrastive pretraining.

## 2 METHODS

Mapping aquaculture from remote sensing data can be framed as a semi-supervised segmentation and classification problem, where given a set of labelled positive and negative examples we would like to segment all water bodies in a region and then classify each segmented water body as an aquaculture pond (pond) or non-aquaculture pond (non-pond) water body. As previous work has successfully identified and segmented water bodies using variants of connected component segmentation, our work primarily focuses on the classification task. As a classification baseline, we trained a random forest using the best performing feature set from previous work. For the baseline deep learning model, we used a standard U-Net. We then tested several modifications to improve both performance and generalization in a low-data setting, including using percentile data to provide temporal information, using variants of masking to incorporate segmentation information, and using contrastive pretraining to harness large volumes of unlabeled data.

### 2.1 Data

While we plan to eventually map aquaculture across the Amazon basin, labelled data was only available for the Brazilian states of Rondônia and Amazonas (see Fig. 1). For positive examples, we received 2322 manually identified aquaculture farms from the Brazilian Agricultural Research Corporation (Embrapa), with 2068 in Rondônia and 254 in Amazonas. Brazilian states are subdivided into municipalities which can naturally be used to spatially split the data. For negative examples, we obtained 2812 examples from five non-aquaculture usage categories from a government dataset, including 848 examples in Rondônia and 1964 in Amazonas [11]. These examples posed several challenges including heavy class imbalances at a state-level, with over twice as many positive examples in Rondônia and nearly 8 times as many negative examples in Amazonas, and further spatial imbalance within each state, where of the 52 municipalities within Rondônia only 7 contained both positive and negative examples.

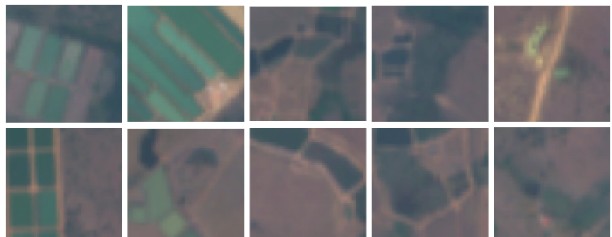

**Figure 2: Examples of aquaculture ponds from Rondônia state. Unlike regions in Asia where post ponds are manmade with strong geometric shapes, aquaculture ponds in the Amazon include geometric ponds (left), ponds built into segments of streams (center), and converted natural water bodies (right).**

### 2.2 Preprocessing and Image Segmentation

In previous studies, water was identified using a combination of standard water indices, such as the NDWI, and variants of connected component segmentation (CCS). However, we found that some labelled examples did not have any pixels classified as water, even using a low NDWI threshold. To avoid missing aquaculture farms, we trained a random forest to detect water from Sentinel-1 and 2 imagery, obtaining positive examples from the Brazilian National Water and Sanitation Agency (ANA) and Embrapa and negative examples from a land-cover classification dataset. To compare the performance of the random forest water classifier to the NDWI, we tested both methods on a set of individually mapped ponds from the Embrapa dataset that were not used in training the random forest and used a NDWI threshold of 0.1. The random forest and NDWI models performed similarly at the pixel level, correctly classifying 93 and 92% of water pixels respectively. However, when considering the number of water bodies in which at least a single pixel was detected, the random forest showed a 4% improvement over the NDWI, detecting some water in 84% of water bodies compared to 80% for the NDWI.

The water-identification random forest was applied across both states to create a binary raster of water pixels and water bodies were extracted using CCS, filtering out all water bodies with fewer than 4 pixels, equivalent to $<400m^2$. Water bodies were then spatially intersected with the annotation polygons and labelled as positive or negative if more than 50% of the water body overlapped the annotation. This resulted in 5485 positive and 2516 negative water bodies in Rondônia state and 284 positive and 1964 negative water bodies in Amazonas state. Due to the heavy class imbalance in Amazonas, negative water bodies were filtered to match the number of positive examples, where we chose the negative water bodies with the shortest distance to a positive example.

Features for the random forest were extracted in Google Earth Engine (GEE) using the workflow described in [44]. For the U-Net, we also used GEE [14] to extract raw Sentinel-1 and Sentinel-2 images; specifically, we extracted 14 spectral bands: the VV band from Sentinel-1 and all bands from Sentinel-2 except for the QA bands. We used the standard GEE preprocessing to filter cloudy pixels. As the U-Net requires a standard input size, for the deep models we cut the satellite imagery into $32 \times 32$ pixel tiles (320 $\times$

320 m$^2$). For each annotated water body, we determined the set of overlapping tiles, considering all tiles with at least 10% annotation or the tile with the most annotation pixels if no tile contained ≥10% annotation. For a water body that appeared in multiple tiles, its classification was calculated as the area-weighted average of classifications from its component tiles.

As labelled examples are only available for two Brazilian states but we hope to map aquaculture across the Amazon, it was important to test generalization to unseen regions. We trained three models, where two were trained on data from single municipalities (Ariquemes and Jaru) and one was trained on a pair of nearby municipalities (Machadinho and Itapuã do Oeste) so that each model had a significant number of positive and negative examples and 300-400 total annotation examples. We validated these models on another municipality with a significant number of positive and negative examples (Monte Negro), and then tested the models on all unseen examples, including data from an additional 17 Rondônian municipalities. We further tested the generalization of the models on examples from the state of Amazonas. As small datasets can lead to variable performance, for each training set and model configuration, models were trained using five random seeds and both the mean and standard deviation in performance is reported.

## 2.3 Random Forest

As a baseline, we trained a random forest classifier using the feature set from [44], which achieved the highest accuracy of previous work. Ten features were extracted at 7 NDWI thresholds, where roughly half of the features are geometric including area, perimeter, compactness and the remaining features take the average of bands or index across the water body. In addition to the 7 NDWI levels, we calculated the features for the water bodies detected by the random forest, as some water bodies had no NDWI signal. For each model, the best random forest was selected from a grid of three tree depths and number of tree parameter combinations using Shannon information gain for rule selection.

## 2.4 Convolutional Neural Network Baseline

All deep models used a U-Net consisting of four up and down double-convolution blocks followed by a fully connected layer. All models augmented images during training through flips and rotations. The U-Net takes in tiles, $\mathbf{X}_i \in \mathbb{R}^{C \times H \times W}$, and outputs a prediction for each tile, $y_i \in \{0, 1\}$. We tested two baselines for the CNN. First, to test the performance of the model without temporal information, we took the median cloud-free value for each channel of each pixel in each $32 \times 32$ tile. Second, to test the performance of the model without radar and multispectral data, we trained a model just using the RGB bands, including temporal information through providing data from multiple percentiles, as described in the next section.

## 2.5 Percentile Data

Previously, hard negative examples like salt pans and rice paddies had been separated from aquaculture ponds through manual inspection of time series imagery. Further, while it may be hard to distinguish a natural pond converted for aquaculture from an unaltered natural pond through a single image, you may expect different patterns over time, such as more consistent water levels in

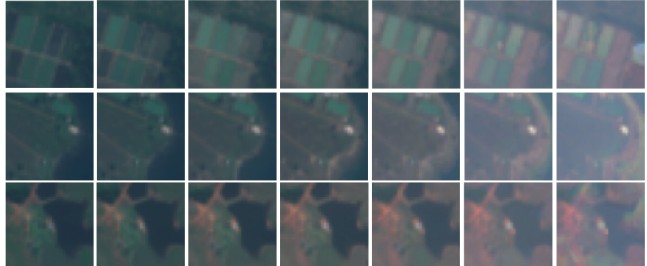

**Figure 3: The percentile data helps capture temporal information, including drained ponds or color changes (top, center) and consistent water levels compared to natural water bodies (center, bottom). Columns show percentiles, increasing from left to right: 5, 10, 30, 50, 70, 90, and 95.**

aquaculture ponds in dry periods. This inspired us to incorporate temporal information into the model. As we have only a few thousand labelled examples and, to test generalizability, we were limited to training on only hundreds of labelled examples, we likely cannot fit the large number of parameters found in more complex architectures like Transformers or provide full satellite time series to the model, as has been done in previous work [39, 45]. Instead, to provide temporal information with a more compact representation, we provided the information for all 14 multispectral channels at seven percentiles: 5, 10, 30, 50, 70, 90, 95. We simply stack the percentiles along the channel dimension, resulting in $14 \times 7 = 98$ channels. Unlike a time series, each percentile level does not represent an image at a particular point in time but instead captures pixel-by-pixel the distribution of values in each channel over time. In particular, the extreme percentiles help capture information about outlier events, such as aquaculture ponds being drained for harvest, without requiring information from a large number of observations (see Fig. 3).

## 2.6 Masked Models

Due to the small number of spatially balanced labelled examples, the U-Net could easily learn features that are incidentally correlated with aquaculture ponds or non-pond water bodies, such as surrounding vegetation. However, these features are not reliable predictors. Further, the baseline classification model does not make use of the segmentation or water detection information. To encourage the model to learn relevant features from the water bodies, we tested three methods of masking: a masked input model where all channels of pixels outside of the segmented area were set to zero in the input, masked average pooling where in the final pooling step before classification information was only averaged over segmented pixels, and masked attention pooling, where instead of taking the average over all segmented pixels, the model learned an attention vector which is masked to the segmented area, allowing some pixels to contribute more to the classification.

While several forms of attention pooling have been suggested [30, 32, 40], we follow the model proposed in [40]. In standard attention, the input $X \in \mathbb{R}^{N \times d}$ is a matrix that contains a $d$-dimensional vector for each of $N$ tokens (assuming a batch size of one for simplicity). In our case, there would be a token for each pixel, $N = 32 \times 32$, where

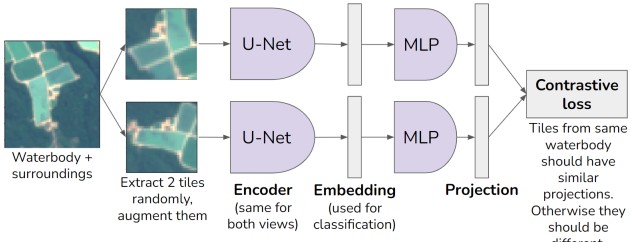

Waterbody + surroundings — Extract 2 tiles randomly, augment them — Encoder (same for both views) — Embedding (used for classification) — Projection — Contrastive loss — Tiles from same waterbody should have similar projections. Otherwise they should be different.

**Figure 4: In contrastive pretraining, each small tile is an augmented view extracted from the same waterbody, but with different position, orientation, and brightness; the model should learn that these are semantically the same.**

each token would have a vector of size $d = 64$ from the output of the final double up-convolution block in the U-Net. $X$ is projected using learned weight matrices $W^Q, W^K, W^V$ into query, key, and value matrices: $Q = XW^Q, K = XW^K, V = XW^V$. To compute attention scores between each pair of tokens, we compute the dot product of each query with each key, followed by a softmax. The softmax scores are used as weights to compute weighted sums over the tokens' value vectors:

$$\text{Attention}(X) = \text{Softmax}\left(\frac{QK^T}{\sqrt{d_k}}\right)V,$$

where $d_k$ is a scaling factor based on the number of attention heads. However, this results in an output with the same dimension as the input and for pooling the dimension of the output must be reduced to a single value. Thus, instead of a query for each token in the input, we learn a single query vector, $q_{cls}$:

$$\text{Attention Pool}(X) = \text{Softmax}\left(\frac{q_{cls}K^T}{\sqrt{d_k}}\right)V.$$

To incorporate segmentation information, we introduce masked attention pooling in which the non-water pixels are set to $-\infty$ before the softmax step, by element-wise multiplying a mask vector, $m$:

$$\text{Masked Attention Pool}(X) = \text{Softmax}\left(\frac{q_{cls}K^T}{\sqrt{d_k}} \odot m\right)V$$

## 2.7 Contrastive Pretraining

While labelled data is scarce, there are numerous unlabeled water bodies from the segmentation step, with spatial and class distributions closer to the true underlying distribution. We were interested in whether this large volume of unlabeled data could be leveraged to learn better low-dimensional representations of complex multi-spectral percentile satellite images. A popular approach that has been very successful in computer vision is *contrastive learning* [4]. This approach extracts two noisy views from each image. Views coming from the same image are encouraged to have similar representations, while being different from the representations of other images' views. The augmentations used to add noise to the data are key to learning a good representation. For example, if images are simply augmented by rotation or flipping, the model can learn

to match histograms of pixel values instead of learning more generalizable representations. While this can be combatted by adding Gaussian noise, adding too much noise can obscure key features.

Most work on contrastive learning has fine-tuned their augmentations for RGB images, while satellite images are multi-spectral and contain many bands outside of RGB. It is unclear how to adapt RGB-specific augmentations like color jitter [4] to work with images with many channels and percentiles. Even if they can be adapted, it is not obvious whether the augmentations used for RGB images will generalize well to non-RGB bands. While some papers have tried to generalize contrastive learning to multi-spectral remote sensing images, they typically rely on geographic proximity to generate views, encouraging tiles that are geographically-close (potentially from different times) to have similar representations [3, 16, 17, 21]. Another approach creates views by extracting different subsets of channels from each image [37]. While these approaches perform well at general tasks like land cover classification, they perform less well on our task. This may be due to the fact that land cover classification relies primarily on the absolute intensity values of each pixel, which is similar for nearby regions. However, our task focuses on a single land cover type (water), and requires classifying based on shape, structure, or subtle differences.

Unlike prior work, we want our model to specifically focus on water bodies, not general landscape characteristics. Specifically, we want two tiles drawn from the *same water body* subject to random augmentations to have similar representations, while tiles from different water bodies should have different representations. Note that labels are not used in contrastive pretraining, so we can pretrain on both labeled and unlabeled data. We pretrain on all the waterbodies in Rondônia and Amazonas within our dataset.

For each distinct water body, we compute a bounding box around it including all segmented water pixels, plus 16 pixels of padding on each side. We extract the tile of satellite imagery, **X**, from that bounding box, which can have arbitrary size, although if it is larger than $256 \times 256$ pixels, we cut it into smaller pieces. During pre-training, we randomly sample two $32 \times 32$ sub-tiles from this bounding box that contain at least 10% water. We then pass both tiles through a large variety of random augmentations, including:

- Random flip/rotate/resize
- Percentile shift: for example, replace the 30th percentile image with a weighted combination of the 30th and 50th percentile images, etc.
- Random sensor drop: inspired by [43], either keep all bands, or zero out one of the following:
  - radar (Sentinel-1)
  - optical bands (Sentinel-2 bands 1-7)
  - infrared bands (Sentinel-2 bands 8-12)
- Random percentile drop
- Random solarize, sharpness
- Mask land: with prob. 0.5, zero out non-water pixels
- Random band-wise linear transformation: for each band, multiply by a random constant in $[0.8, 1.2]$ and add a random constant in $[-0.2, 0.2]$.

After passing both sub-tiles through these augmentations, we have two strongly distorted "views" of the same water body, $\tilde{\mathbf{X}}_i$ and $\tilde{\mathbf{X}}_j$. We then follow the SimCLR framework proposed by [4].

**Table 1: Performance metrics for all models trained on municipalities in Rondônia and tested on unseen municipalities within Rondônia. Values show mean and standard deviations over three spatial splits and five random seeds. Note that for contrastive learning, we only pretrained one model due to computational constraints; we instead used 5 different seeds to train the linear classifier. Bold values indicate the best mean performance across models. RF = random forest.**

| | | U-Net Models | | | | | | | |
|---|---|---|---|---|---|---|---|---|---|
| | RF | Median | RGB | Percentile | Masked Input | Masked Pool | Attn. Pool | Masked Attn. Pool | Contrastive |
| Prec. | 0.74 ± 0.01 | 0.89 ± 0.00 | 0.88 ± 0.01 | **0.95 ± 0.01** | 0.86 ± 0.02 | 0.92 ± 0.02 | 0.93 ± 0.03 | **0.95 ± 0.01** | 0.91 ± 0.01 |
| Rec. | **0.99 ± 0.01** | 0.97 ± 0.01 | 0.94 ± 0.01 | 0.94 ± 0.01 | 0.98 ± 0.01 | 0.98 ± 0.01 | 0.93 ± 0.04 | 0.95 ± 0.02 | 0.95 ± 0.02 |
| F1 | 0.84 ± 0.01 | 0.92 ± 0.01 | 0.91 ± 0.00 | **0.95 ± 0.01** | 0.92 ± 0.01 | **0.95 ± 0.01** | 0.93 ± 0.04 | **0.95 ± 0.01** | 0.93 ± 0.01 |
| Acc. | 0.77 ± 0.01 | 0.89 ± 0.01 | 0.87 ± 0.01 | **0.93 ± 0.01** | 0.87 ± 0.02 | **0.93 ± 0.02** | 0.90 ± 0.02 | **0.93 ± 0.01** | 0.90 ± 0.01 |

**Table 2: Performance metrics for models trained on data from Rondônia on Amazonas state. Values show mean and standard deviations over three spatial splits and five random seeds. Bold values indicate the best mean performance across models. RF = Random forest.**

| | | U-Net Models | | | | | | | |
|---|---|---|---|---|---|---|---|---|---|
| | RF | Median | RGB | Percentile | Masked Input | Masked Pool | Attn. Pool | Masked Attn. Pool | Contrastive |
| Prec. | 0.34 ± 0.32 | 0.97 ± 0.01 | 0.91 ± 0.01 | **0.99 ± 0.01** | 0.70 ± 0.04 | 0.88 ± 0.05 | 0.96 ± 0.03 | 0.89 ± 0.02 | 0.94 ± 0.00 |
| Rec. | 0.40 ± 0.34 | 0.74 ± 0.06 | 0.31 ± 0.03 | 0.56 ± 0.04 | 0.93 ± 0.01 | **0.87 ± 0.03** | 0.55 ± 0.04 | 0.86 ± 0.05 | 0.85 ± 0.01 |
| F1 | 0.35 ± 0.30 | 0.81 ± 0.06 | 0.44 ± 0.05 | 0.69 ± 0.04 | 0.80 ± 0.02 | 0.87 ± 0.02 | 0.65 ± 0.04 | 0.86 ± 0.02 | **0.89 ± 0.00** |
| Acc. | 0.30 ± 0.29 | 0.86 ± 0.03 | 0.64 ± 0.02 | 0.78 ± 0.02 | 0.76 ± 0.04 | 0.86 ± 0.03 | 0.77 ± 0.02 | 0.86 ± 0.01 | **0.89 ± 0.00** |

We pass both views through a U-Net encoder, $f_{enc}(\cdot)$, to obtain embeddings of each view, $\mathbf{h}_i, \mathbf{h}_j \in \mathbb{R}^d$, using embeddings of size $d = 256$. We further pass these embeddings through a MLP projection head, $g(\cdot)$, to obtain projection vectors $\mathbf{z}_i = g(\mathbf{h}_i)$. We then apply the NT-Xent loss function [4] on a batch of $2N$ projection vectors from $N$ water bodies. For a given view, we want its embedding to be similar to that of the other view of the same water body, while being different from all other tiles in the batch. For a given water body's two augmented views, $(i, j)$, the loss is defined as

$$\ell_{i,j} = -\log \frac{\exp(sim(\mathbf{z}_i, \mathbf{z}_j)/\tau)}{\sum_{k=1}^{2N} \mathbb{1}_{[k \neq i]} \exp(sim(\mathbf{z}_i, \mathbf{z}_k)/\tau)},$$

where $\tau = 0.07$ is a temperature parameter, and the similarity function $sim(\mathbf{u}, \mathbf{v}) = \mathbf{u}^T \mathbf{v}/(\|\mathbf{u}\|\|\mathbf{v}\|)$ is the dot product between the L2-normalized embeddings.

After pre-training is complete, we freeze the encoder, use it to extract embeddings from each tile, and then train a linear classifier on top of these fixed embeddings. We also tried fine-tuning the encoder, but this did not improve results, so we report results for the fixed embedding version.

## 2.8 Training Configuration

Data was prepossessed and spatially split as described in Sec. 2.2. For all CNN models we used the RMSProp optimizer and tried learning rates from {1e-2, 1e-3, 1e-4, 1e-5, 1e-6}. We chose the learning rate based on F1 on the validation municipality, Monte Negro. Across all CNN models, 1e-4 was consistently the best performing learning rate. We used a batch size of 16, and found that 20 epochs was sufficient for convergence. For contrastive learning, for pretraining we used a learning rate of 3e-4 and a batch size of 512, stopping when the loss did not improve for 10 epochs. While contrastive learning generally requires pretraining with large batches [4], we used ghost normalization [5] to retain the regularizing effect of batch normalization with small batches. For training the linear classifier we also used 20 epochs, batch size 16, and tried learning rates {1e-2, 1e-3, 1e-4, 1e-5, 1e-6}, again using the validation municipality to choose the learning rate.

## 3 RESULTS

Previous work primarily used accuracy to evaluate aquaculture classification performance. However, the F1 score can better reflect performance on imbalanced data:

$$\text{Prec} = \frac{TP}{TP + FP}, \quad \text{Rec} = \frac{TP}{TP + FN}, \quad \text{F1} = \frac{2 \, \text{Prec} \cdot \text{Rec}}{\text{Prec} + \text{Rec}}$$

Further, precision and recall alone provide important information on model performance. For example, it may be preferable to choose a model with high recall that does not misclassify any aquaculture ponds (TPs) and use post-processing steps to filter false positives (FPs). Precision, recall, F1, and accuracy scores for all models on unseen data in Rondônia state are reported in Table 1.

The random forest performed significantly worse than the baseline deep learning models, achieving 10% lower accuracy and 7% lower F1 score. Adding the percentile data improved the accuracy a further 4% and increased the F1 score to 0.95, with all deep models achieving a low standard deviation of 0.01 for both metrics across the three spatial splits and five random seeds. Masking the input data decreased performance relative to the percentile data by 5% in accuracy and 3% for F1. However, masking through masked pooling and masked attention pooling both achieved the same performance as the percentile model. Adding attention pooling without masking decreased performance by 2-3% on all metrics. Contrastive pretraining also decreased performance by 2% F1 and 3% accuracy.

Performance generally decreased significantly when tested on examples from the state of Amazonas (see Table 2). The random

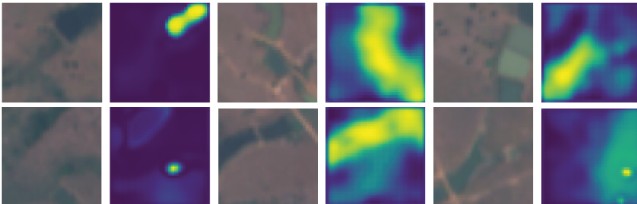 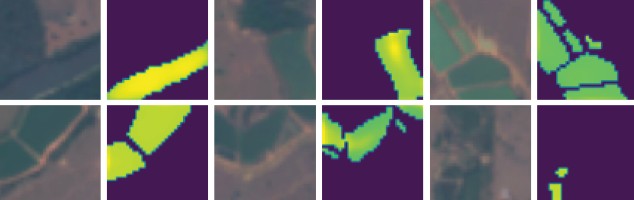

**Figure 5: Examples of pairs of RGB images (left) and attention vectors (right). While in general the attention vectors put heavy weight on the water bodies, they also often put weight on portions of the surrounding landscape, potentially learning incidentally correlated features that hurt generalization.**

**Figure 6: Examples of pairs of RGB images (left) and masked attention vectors (right). The masked attention vectors are generally uniform, suggesting masked attention pooling mostly benefits from the masking of surrounding areas.**

forest predicted most water bodies but also predicted some false negatives, leading to performance worse than chance. The deep models performed better, but still saw a significant decrease in performance and increase in performance variability. The median model generalized better than just using the percentile data, achieving accuracy and F1 scores of 86% and 0.81 compared to 78% and 0.69 for the percentile model. Masking the input or using attention pooling resulted in a 1-2% decrease in accuracy compared to the percentile data. However, the masked pooling and masked attention pooling models achieved the same accuracies as the median model and a 5% better F1 score with lower variability. Contrastive pretraining resulted in a slight further improvement (2-3%), with an accuracy and F1 of 89%.

## 4 DISCUSSION

The previous hand-crafted feature set used to detect aquaculture ponds in Asia generalized poorly to the Amazon basin. While in Asia random forests trained on the feature set achieved accuracies over 95%, in the Amazon they achieved only 77% accuracy and resulted in models that generalized worse than chance. One potential reason for the poor performance is that many aquaculture ponds in Asia are man-made with clear geometric shapes, making them easy to distinguish from natural water bodies using features like area-to-perimeter ratio. However, in the Amazon there is more heterogeneity in aquaculture ponds, including converted natural ponds and ponds built from sectioned off stream segments. Another potential reason for the random forests' poor performance is that some water bodies in the Amazon did not have a strong NDWI signal, leading to minimal features. It is unclear if ponds with a poor NDWI signal are specific to the region, or if prior methods were potentially missing aquaculture ponds with weak NDWI signals. While it may be possible to design a better feature set for the Amazon, the deep learning models perform well without the need for extensive feature engineering.

Incorporating temporal information through percentile data improved metrics by 3-4% within Rondônia but alone led to poorer generalization than the median model. A potential reason for the poorer generalization could be high levels of cloud-cover in Amazonas. Even with filtering during preprocessing, in some cases cloud pixels appeared in the high percentiles and these tiles were often misclassified as ponds. While the median model generalized well,

the RGB model generalized extremely poorly with over a 45% drop in F1 and a 25% drop in accuracy. These results suggest that data beyond RGB bands, such as multispectral and radar data is important for learning robust models.

Masked pooling and masked attention pooling achieved the best performance in Rondônia and significantly improved performance in Amazonas over the standard percentile model, only generalizing 2% worse than the contrastive model. In contrast, standard attention pooling decreased performance compared to simply using percentile data. Visualizing example attention vectors, standard attention pooling may perform slightly worse because it often places weight broadly outside of the water body, potentially learning correlated features such as vegetation (see Fig. 5). The masked attention vectors show fairly uniform weight across water bodies, suggesting that the masked attention pooling is not learning to weight certain areas' pixels, such as boundaries, more strongly than others (see Fig. 6). Thus, its performance was similar to the simpler masked pooling approach, which simply averages the feature vectors across the water pixels. These results suggest that it can be beneficial to incorporate segmentation information through masking. However, masking the input also performed worse than simply using the percentile data. The benefit of masked pooling over the masked input model may be that masking at the last step in the network allows the representations for the water pixels to incorporate information from the surrounding land pixels, allowing for contextual information without overfitting.

Contrastive pretraining succeeded in extracting reasonable features from the data, where training a linear classifier on top of the extracted fixed embeddings achieved 93% F1 on Rondônia and 89% F1 on Amazonas. This performance is still 2% behind the supervised percentile model and the masked pooling models in Rondônia. However, it performs the best out of all methods when generalizing to Amazonas, achieving 2% better performance on both F1 and accuracy than the masked pooling methods and significantly better than the supervised percentile model. Surprisingly, fine-tuning did not improve results. While not shown in the table, the contrastive learning results varied significantly based on the exact augmentations used, and even the learning rate used to train the linear classifier. Thus, even though contrastive learning has potential to improve generalization, additional work is needed to make it more reliable.

Both the contrastive learning and masked pooling approaches encourage the model to focus on water, by only allowing pooling

over water pixels or by masking out non-water pixels randomly in the contrastive augmentations respectively, but do not completely prevent the model from learning contextual information. We hypothesize that this is a key factor why both approaches generalize much better to Amazonas than the percentile model. The band-wise linear transformation augmentation also forces the model to pay more attention to relative shapes instead of absolute pixel values, as multiplying a band by a constant would modify the absolute values, but retain the rough shape of the pond, which may matter more. This augmentation may be useful in other contexts to help models focus on shape over texture, which has been noted as a challenge for CNNs [12, 15].

While Rondônia and Amazonas are both states within Brazil, they cover more than $1,000,000\ km^2$ - an area more than three times the size of France. The states have significantly different climates, vegetation, and levels of development, as well as significant variation within each state. Thus, the models' ability to perform well in Amazonas state when only trained using data from Rondônia suggest that the methods proposed here can help with generalization to significantly different regions. However, more data and testing would be needed to understand how well these methods would generalize across the Amazon or even to other continents.

In principle, these modifications could be applicable to a range of remote sensing tasks. For example, incorporating temporal information through percentile data would help in crop type mapping, where crops show different growth patterns over time [34]. Masked pooling could help in tasks like building classification [36] and brick kiln detection [20], where some background information could be useful but incidental correlations would also likely occur. However, further experiments would be needed to evaluate the performance of these methods in other scenarios. Further, these methods are isolated to the classification step, allowing them to potentially be incorporated into a variety of segment-and-classify pipelines.

## 5 CONCLUSION

While the availability of large medium-resolution satellite datasets creates the potential for low-cost mapping of a variety of infrastructure, many applications only have access to small numbers of labelled examples that are often spatially biased and include heavy class imbalances. We show that with as few as 300 labelled examples, standard deep learning models can outperform random forests for aquaculture detection by 10% without the need for hand-crafted features. Performance can be further improved by using percentile data to compactly add temporal information. Generalization to unseen regions can be improved by incorporating segmentation information without eliminating contextual information through masked pooling and contrastive pretraining. Contrastive pretraining further allows the model to harness the large quantities of unlabelled data, including from regions lacking labeled examples. Further, both masked pooling and contrastive pretraining can help address the challenge of spatial imbalance, by reducing emphasis on background features and incorporating unlabeled data respectively. While these strategies were applied to detect aquaculture ponds, they could potentially be useful for other tasks with a segment-then-classify workflow, such as solar panel and livestock barn mapping

or building classification. While these methods improved significantly over baseline approaches, there is still room for improvement in generalization to new regions. Further, more work is needed to understand and develop augmentations for multispectral images.

## ACKNOWLEDGMENTS

We thank the Amazon Fund for sponsoring the Strategic Territorial Intelligence System for Aquaculture in the Amazon Project, which allowed manual mapping of the aquaculture ponds. This work was supported by a Schmidt Program Fellowship to F.S.P, a Cornell Atkinson Academic Venture Fund award to A.S.F and C.P.G, a Minas Gerais State Agency for Research and Development award (APQ 02.629/21) to N.O.B. and a São Paulo Research Foundation award (2022/10443-6) to F.S.P, M.E.U, N.O.B, B.R.F and A.S.F, National Science Foundation Award CCF-1522054 to C.P.G, and by AFOSR DURIP grant FA9550-21-1-0316, AFOSR MURI grant FA9550-18-1-0136, and AFOSR grant FA9550-20-1-0421 to C.P.G.

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
