# OpenReview forum: "Detecting Aquaculture with Deep Learning in a Low-Data Setting"
_KDD.org/2023/Workshop/Fragile_Earth — KDD 2023 Workshop Fragile Earth Submission_

### Official Review · Reviewer_jSDw · 2023-07-05
**review of "Detecting Aquaculture with Deep Learning in a Low-Data Setting"**

**Rating:** 7
**Confidence:** 2

**Review:**

Summary:
This paper proposes a method for identifying aquaculture facilities using deep learning techniques and satellite imagery. The method is designed to work effectively even in situations where there is limited data available. The paper discusses the use of contrastive learning, a technique that has been successful in computer vision, and the challenges of adapting it for multi-spectral remote sensing images.

Strengths
1)	The paper presents a novel application of deep learning techniques for detecting aquaculture facilities using satellite imagery. This is a significant contribution to the field of remote sensing and environmental science.
2)	The research addresses a real-world problem of detecting aquaculture facilities, which has implications for environmental management and policy.

Weaknesses
1)	The paper does not provide a clear explanation of the methodology used in the research. More details about the training process, such as the data used for training and the specific steps taken during training, would be beneficial.

Questions
1)	How does the model deal with the heavy class imbalances at a state-level and further spatial imbalance within each state?
2)	How scalable is the proposed method? Can it handle large volumes of data from different regions?
3)	How can the proposed method be integrated into existing systems and processes for environmental management and policy?

---

### Official Review · Reviewer_it8a · 2023-07-12
**This is an excellent empirical paper addressing the problem of detecting aquaculture with deep learning in low-data settings.**

**Rating:** 8
**Confidence:** 4

**Review:**

This paper addresses the problem of detecting aquaculture in low-data settings, using deep learning techniques and open-source medium resolution satellite imagery data. The paper begins by providing a motivation for tackling this problem by articulating the need to map out detailed information on acuaculture farms in the world in order to promote the optimal practice of acuaculture as it can potentially improve the globe's overall carbon footprints significantly. The paper then outlines the challenges in doing so given the low-data availability of open-source satellite imagery data and their issues (few labels, bias, etc.), particularly in the Amazon region where the characteristics of acuacultural farms are different from Asia where the previous studies have focused on and the data are scarce. The paper proposes a number of techniques to enhance and adapt the deep learning (U-Net) approach to this problem, ranging from incorporating temporal information via percentile data, use of masked models to contrastive pre-training. Their experiments demonstrate that through these techniques, they significantly manage to out-perform the accuracy of baseline models. They also show the "generalizability" of their models (particularly those trained with contrastive pre-training) by testing them across distinct manucipalities. The exposition is well-written and clear in general, and the experimental section provides details making the work reproducible. Overall this is an excellent paper, addressing an important real world problem towards sustainable acuacultural practices, while addressing topical technical agenda in the application of upto date deep learning technologies.

---

### Official Review · Reviewer_UVg8 · 2023-07-13
**Review of "Detecting Aquaculture with Deep Learning in a Low-Data Setting"**

**Rating:** 8
**Confidence:** 4

**Review:**

This paper is focused on using deep learning techniques to detect aquaculture locations when labelled data is sparsely available (as little as 300 samples). Overall, the paper is well written and authors back their claims with experiments. The paper has two parts, a. segmentation, b. classification. It starts with segmenting water bodies followed by using a classifier to determine the segmented group of pixels' class. I especially liked the experiments section where each design decisions were shown to illustrate their impact. I think the paper is more generalizable than only aquacultural lands (which is actually good). Therefore, I expect authors to add a couple of discussion points regarding this. In addition, I understand that the labelled training samples may not always be available for "a similar" region, and we are addressing this. How about completely different regions? Is there a way to illustrate/show the performance impact on these? Finally, the paper addresses one of the themes of the workshop and can address a key concept of "sustainability".

---

### Official Review · Reviewer_gQAK · 2023-07-14
**Review for "Detecting Aquaculture with Deep Learning in a Low-Data Setting"**

**Rating:** 8
**Confidence:** 2

**Review:**

Summary : Using Deep learning to find aquaculture location is given in the low data scenario. Using Semi-supervised learning and combining temporal data using percentile data , method of masked pooling and contrastive learning,  they have shown significant performance increase.

Strengths:
-It is a well written paper, problem, data , method and results have been clearly presented.
-Temporal data has been accounted for.
-Results are promising and discussion into limitations and future work has been provided.

Weaknesses :
- Details on performance metrics isn't provided.

---

### Official Review · Reviewer_Gz6c · 2023-07-16
**Review of "Detecting Aquaculture with Deep Learning in a Low-Data Setting"**

**Rating:** 8
**Confidence:** 4

**Review:**

Summary:

This paper incorporates temporal information through percentile data and shows that deep learning models can outperform previous methods by as much as 15% with as few as 300 labelled examples. Moreover, the authors show that generalization to unseen
regions can be improved by incorporating segmentation information through masked pooling and using contrastive pretraining to
harness large quantities of unlabelled data.

Strengths:
- The paper is clearly written.
- The authors provide detailed description of the contrastive pretraining procedure.
- Good performance are obtained according to experiments.

Weaknesses:
- Some prior works can be moved to the introduction section.
- The title of the section 2.4 'CNN Baseline' should be 'Convolutional Neural Networks Baseline'.

---

### Decision · Program_Chairs · 2023-07-19

**Decision:**

Accept (Oral)

**Comment:**

Congratulations!

We are pleased to inform you that your submission: Detecting Aquaculture with Deep Learning in a Low-Data Setting has been accepted to The KDD 2023 Workshop Fragile Earth: AI for Climate Sustainability - from Wildfire Disaster Management to Public Health and Beyond.

Camera ready deadline is ** July 24 AOE **.  Please log in to OpenReview and prepare your camera-ready version based on the reviews. Formatting rules are the same as for the initial submission and submissions must adhere to KDD 2023 guidelines available at https://authors.acm.org/proceedings/production-information/taps-production-workflow.

Again, congratulations on the acceptance of your paper!  We look forward to seeing you at the workshop on Aug 7, 2023.

The Fragile Earth Workshop Proceeding Chairs